# Long-term effects of reovirus strain T3D on the myocardium

Maryam Ebadi Fard Azar,[1] Marcelle Dina Zita,[1] Kshipra S. Keole,[1] Sylvie Rousseau,[1] Charles Cohen,[1] Dao-Fu Dai,[2] Karl W. Boehme,[3] Luigi Adamo[1]

**ABSTRACT** Mammalian orthoreovirus (reovirus) is a well-established model for studying viral pathogenesis. Although studies of reovirus have largely been focused on central nervous system disease, reoviruses can also cause myocarditis. Reovirus strain type 1 Lang (T1L) causes mild myocarditis in neonatal mice. However, many highly myocarditic reoviruses are reassortants between T1L and the serotype 3 (T3) Dearing strain that is non-myocarditic. Although cardiac immune responses to T1L are well described, many open questions remain regarding cardiac immune responses to T3 reoviruses. To better understand the effects of T3 reoviruses on the heart, we investigated the long-term cardiac impact of reovirus strain T3 Dearing (T3D) in neonatal C57BL/6 mice. Oral infection with T3D resulted in subclinical myocarditis that was non-lethal but still produced persistent histological myocardial alterations. Echocardiographic analysis revealed a mild decrease in diastolic left ventricular anterior wall thickness in mice infected with $10^4$ PFU, though no consistent dose-dependent functional impairments were observed. Histological examination identified myocardial lesions characterized as replacement fibrosis that developed independently of the inoculating dose. Flow cytometry showed an early immune response at 8 days post-infection, with increased CD4 T cells, CD8 T cells, B cells, and innate immune cells. By 26 days post-infection, the inflammation had largely resolved, but low-level immune infiltration persisted, characterized by CD4 T cells, CD8 T cells, and B cells. These findings suggest that T3D induces subclinical myocarditis with lasting histopathological changes. The presence of fibrosis raises concerns about potential long-term cardiac effects, emphasizing the need for further research into the myocardial impact of non-myocarditic reoviruses.

**IMPORTANCE** Viral infections that cause myocarditis are a significant cause of morbidity and mortality worldwide, particularly in children and young adults. Mammalian orthoreoviruses (reoviruses) are an established model for studying viral myocarditis in mice and are also under development as a cancer therapeutic due to their capacity to kill cancer cells. Here, we describe the long-term myocardial effects of the type 3 Dearing (T3D) reovirus strain, which is classically considered neurotropic. Our findings indicate that T3D causes subclinical myocarditis that is non-lethal but produces histopathological changes indicative of fibrosis. Thus, although T3D does not cause overt acute cardiac pathology, it can have long-term effects on the myocardium. This work will inform future studies on reovirus tropism and is relevant to ongoing efforts to harness the oncolytic properties of reoviruses for therapeutic applications.

**KEYWORDS** reovirus, heart, myocarditis

Mammalian orthoreovirus (reovirus) is a double-stranded RNA (dsRNA) virus that infects most people during childhood, but the disease is typically subclinical (1). Reovirus has also been implicated in the breakage of tolerance to dietary antigens, such as gluten, leading to celiac disease (2, 3). The clinical significance of reovirus is growing, as its oncolytic properties have attracted interest for the potential development

Address correspondence to Luigi Adamo, ladamo2@jhmi.edu, or Karl W. Boehme, KWBoehme@uams.edu.

Maryam Ebadi Fard Azar and Marcelle Dina Zita contributed equally to this article. The author order was determined alphabetically

Karl W. Boehme and Luigi Adamo contributed equally to this article.

L.A. is co-founder of i-Cordis, LLC, a startup company focused on the development of immunomodulatory therapies for heart failure, and he is a consultant for Kiniska Pharmaceuticals and Novo-Nordisk. None of the other authors report any conflict of interest.

of oncolytic virotherapeutics (4). It is, therefore, important to expand our understanding of the pathologic effects of reovirus on different organs.

Much of what is known about reovirus pathogenesis has been learned from studies in mice and largely focuses on disease caused in the central nervous system (CNS). Three reovirus serotypes circulate in humans: serotype 1 (T1), serotype 2 (T2), and serotype 3 (T3); T1 and T3 reoviruses are well-studied in laboratory animals (1). T1 and T3 reoviruses differ in the route of dissemination and the cells targeted in the CNS (5). T1 strains reach the CNS through the bloodstream, targeting ependymal cells and leading to hydrocephalus. In contrast, T3 reoviruses spread to the CNS via a combination of neural and hematogenous pathways, where they infect neurons and cause fatal encephalitis.

However, reoviruses can also cause disease in other organ systems, including the heart, where they can cause myocarditis (6). The prototype reovirus laboratory strain T1 Lang (T1L) is mildly myocarditic, producing non-lethal cardiac inflammation in wild-type mice (1, 7, 8). The myocarditic capacity of T1L can be markedly increased by reassortment with reovirus strain T3 Dearing (T3D), which is non-myocarditic (7, 9, 10). Although numerous studies have examined the effect of T1L and T1L × T3D reassortants in the heart, less is known about the cardiac impact of T3D (1, 7–10). In this study, we performed an in-depth analysis of the long-term effect of T3D on the heart. We found that T3D reovirus causes subclinical myocarditis that progressively resolves but results in persistent histological myocardial alterations.

## RESULTS AND DISCUSSION

To assess the effects of reovirus T3D on cardiac function, neonatal C57BL/6 mice were either mock infected with phosphate-buffered saline (PBS) or orally inoculated with $10^4$, $10^5$, $10^6$, or $10^7$ plaque forming units (PFU) of T3D. At 26 to 35 days post-infection, cardiac function was assessed via echocardiogram (Fig. 1A). At a dose of $10^4$ PFU of T3D, the diastolic left ventricular (LV) anterior wall (LVAW) thickness was significantly decreased in comparison with the control group ($P < 0.05$). However, no difference in LVAW thickness was observed in mice infected with higher viral doses. We did not observe differences in diastolic LV posterior wall thickness (LVPW), LV end diastolic diameter (LVEDD), or LV end systolic diameter (LVESD) between the control group and any of the reovirus-infected groups (Fig. 1B through D). Furthermore, no significant differences in ejection fraction (EF), LV mass, or stroke volume (SV) between the control group and the reovirus-infected mice were observed (Fig. S1A, B, and D). We found a small difference in cardiac output between mock-infected mice and mice infected with T3D at doses of $10^5$ or $10^6$ PFU per mouse ($P < 0.05$) (Fig. S1C). However, none of the differences observed were dose-dependent. These findings indicate that T3D does not impair cardiac function.

We next performed histological analysis of hearts from mice infected with $10^4$, $10^5$, or $10^6$ PFU per mouse of T3D at approximately 28 days post-infection. Masson's trichrome staining of histological sections revealed myocardial lesions in T3D-infected hearts (Fig. 2). However, there was no clear correlation between viral dose and the number of pathological lesions per section. These findings indicate that T3D caused low-grade chronic myocarditis. Notably, the lesions were characterized by a cardiac pathologist as displaying "replacement fibrosis," which indicates the lesions were undergoing repair and healing. Inflammation and fibrotic changes in cardiac tissue can be an underlying cause of more serious cardiac problems. Fibrosis, for instance, is associated with an increased risk of arrhythmias and sudden cardiac death (11). This finding suggests that infection with viruses that do not cause overt cardiac phenotypes might still result in subclinical cardiac damage that might contribute to developing long-term cardiac pathologies.

Next, we assessed the acute cardiac immune responses induced by T3D. Myocarditic reoviruses disseminate to the heart rapidly and can induce myocarditis as early as 8 days post-infection (9, 12). At 8 days post-infection with $10^6$ PFU of T3D, we observed that the total number of CD45+ cells was increased in the T3D-infected hearts compared to controls (Fig. 3A). Among innate immune cells, monocytes (Ly6C+), neutrophils (Ly6G+), and natural killer cells (NK1.1+) were significantly increased in infected mice compared to

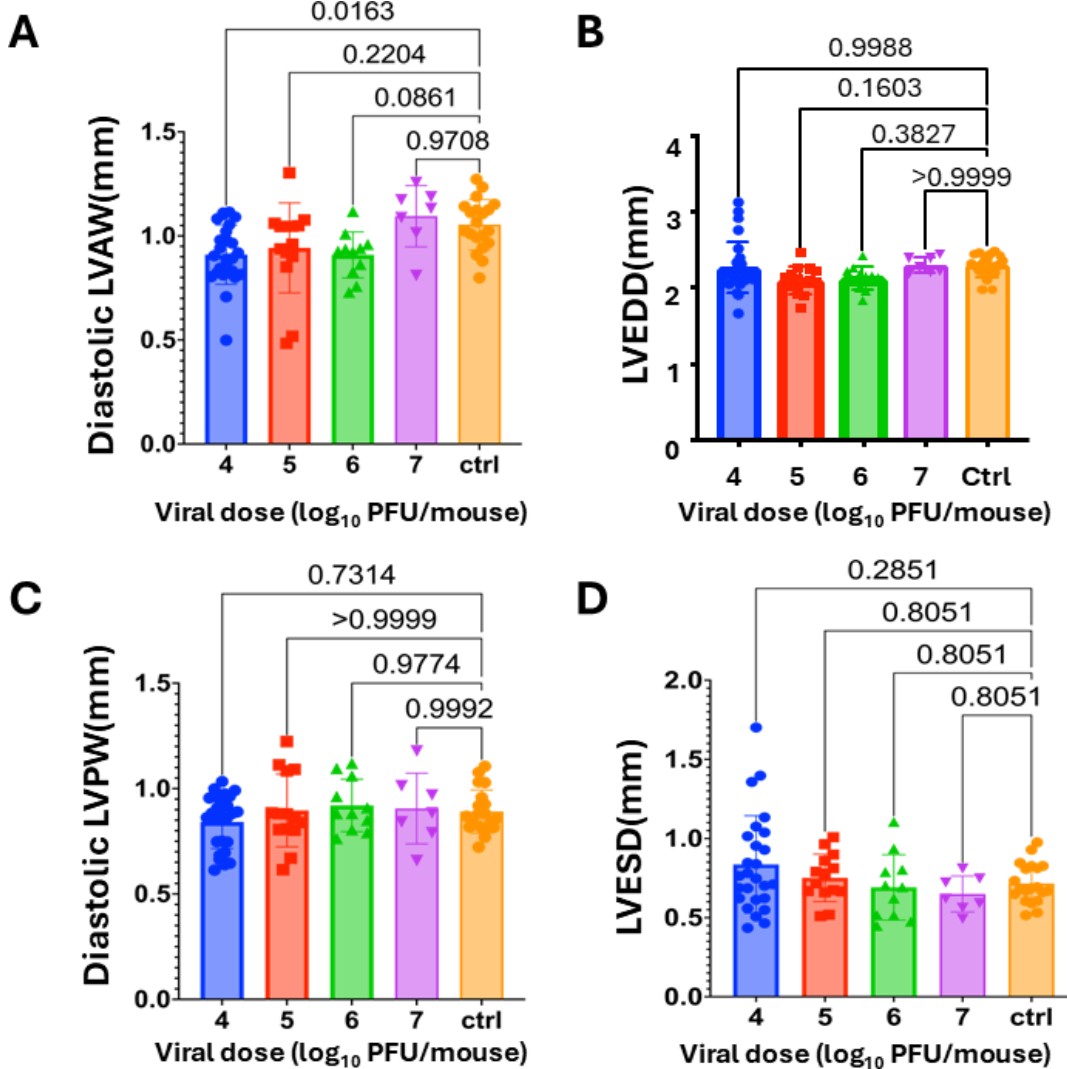

**FIG 1** Echocardiography analysis at 4 weeks post-infection. Neonatal C57BL/6 mice were mock-infected (Ctrl; $N = 20$) or infected orally with $10^4$ ($N = 26$), $10^5$ ($N = 14$), $10^6$ ($N = 11$), or $10^7$ PFU ($N = 7$) of T3D. (A) Diastolic LVAW thickness, (B) diastolic LVPW thickness, (C) LVEDD, and (D) LVESD were measured at 26–35 days post-inoculation. Each dot represents a recording for a given mouse. Each parameter was measured three times in each mouse. The average recording for each mouse is reported. *P*-values represent ANOVA followed by pairwise comparisons, with *P*-values adjusted using the false discovery rate method.

control animals (Fig. 3B). The total number of macrophages (CD64+) was also higher compared to the control group ($P = 0.0519$), albeit not to a statistically significant level (Fig. 3B). Reovirus-infected animals also had significantly higher levels of adaptive immune cells (B cells [CD19+], CD4 T cells, and CD8 T cells) than controls (Fig. 3C). These data indicate that T3D induces acute cardiac inflammation.

In C57BL/6 mice, reovirus is cleared within 20 days (12–17). Histological analysis indicated that at 4 weeks post-infection with T3D, cardiac lesions remained detectable, suggesting that T3D-induced cardiac inflammation persists beyond the time of viral clearance (Fig. 2). To define cardiac inflammation at later time points, we measured cardiac immune infiltration of hearts infected with $10^6$ PFU of T3D at 26 days post-infection (Fig. 4). Cardiac immune responses were characterized by an increase in CD45+ cells ($P = 0.06$), albeit not to statistically significant levels (Fig. 4A). We found fewer innate immune cells in the heart at this time point (Fig. 4B). Monocytes (Ly6C+) were significantly increased relative to controls, while there was no difference in the numbers of macrophages (CD64+) and natural killer cells (NK1.1+). There was also a significant decrease in the total number of neutrophils (Ly6G+). However, significantly more

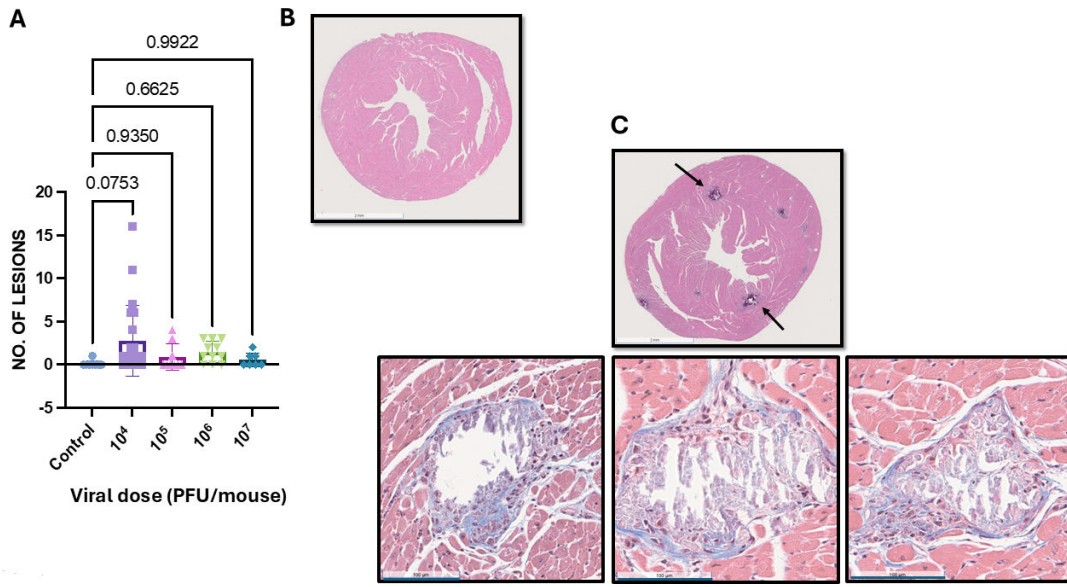

**FIG 2** Myocardial histological analysis at 4 weeks post-infection. Neonatal C57BL/6 mice were mock-infected (Ctrl; N = 8) or infected orally with $10^4$ (N = 22), $10^5$ (N = 9), $10^6$ (N = 11), or $10^7$ (N = 7) PFU of T3D. At 4 weeks post-infection, histological analysis was performed using Masson's trichrome staining. (A) The number of histological lesions per myocardial mid-papillary section was quantified for each dose of T3D. Each dot represents the number of lesions identified in each mouse analyzing a mid-papillary ventricular section. P-values represent ANOVA followed by pairwise comparisons, with P-values adjusted using the false discovery rate method. (B) Representative image for mock-infected control tissue. (C) Representative images are shown for (left panel) $10^4$, (middle panel) $10^5$, or (right panel) $10^6$ PFU of T3D. Scale bars reported at the bottom left corner of each image = 100 µM. A wider field representative image is shown to depict the entire tissue.

adaptive immune cells, B cells (CD19+), CD4 T cells, and CD8 T cells (Fig. 4C), were detected in T3D-infected mice compared to controls. It is not clear why the immune infiltration persists at this time point. The result may indicate that reovirus antigen has not been completely cleared by this time point. Alternatively, the sustained immune response could result from a response to self-antigens, which occurs with coxsackievirus (18, 19). Reovirus infection of the biliary tract is associated with autoantibody production (20), and the same effect may occur in the heart.

In this study, we found that reovirus strain T3D causes myocarditis in early post-natal mice. We observed that T3D infection in newborn mice causes a strong inflammatory infiltration early post-infection that evolves into chronic low-level inflammation and fibrosis. This work may have implications for human trials using T3D as an oncolytic agent, as the adaptive immune response to the virus may still produce subclinical cardiac damage that could contribute to long-term cardiac pathology.

## MATERIALS AND METHODS

### Cells

Spinner-adapted murine L929 fibroblasts were maintained in Joklik's modified minimum essential medium (Sigma) supplemented to contain 5% heat-inactivated fetal bovine serum (FBS; Invitrogen), 2 mM L-glutamine (Invitrogen), 100 U/mL penicillin-100 µg/mL streptomycin (Invitrogen), and 25 µg/mL amphotericin B (Sigma).

### Viruses

Reovirus strain T3D (Fields) was generated using plasmid-based reverse genetics. Reovirus virions were purified from second- or third-passage L929 cell lysates infected with twice-plaque-purified reovirus. Reovirus particles were Vertrel (TMC Industries) extracted from infected-cell lysates, layered onto 1.2 to 1.4 g/cm$^3$ CsCl gradients, and

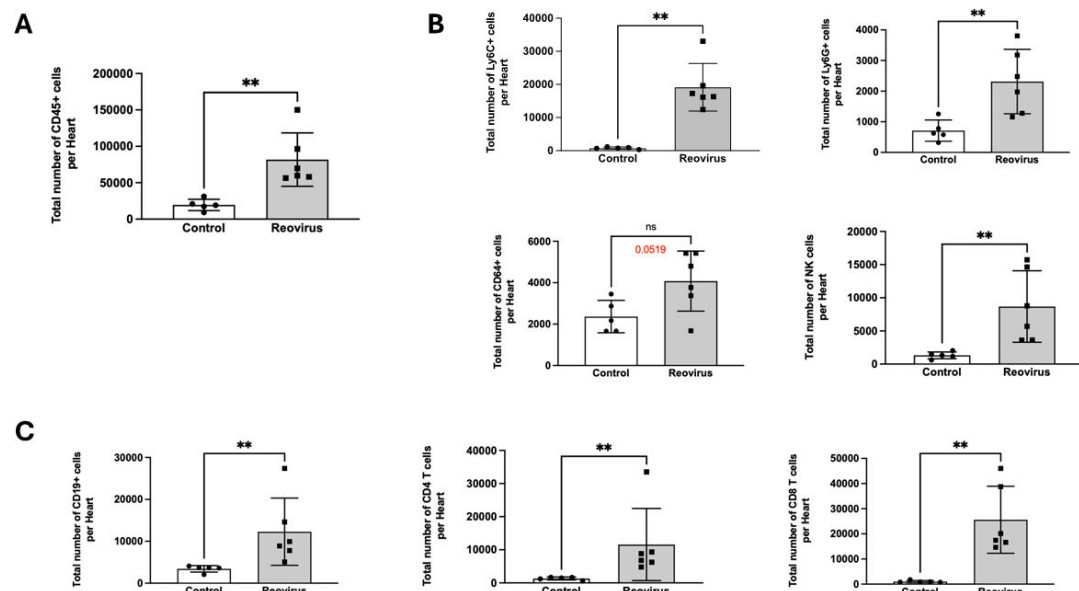

**FIG 3** T3D induces acute myocardial immune infiltration. Neonatal C57BL/6 mice were mock-infected (Control; $N$ = 5) or infected orally with $10^6$ PFU of T3D ($N$ = 6). At 8 days post-infection, the total number of cells per heart was quantified via flow cytometry focusing on the following cell types: (A) CD45+ cells, (B) innate immune cells (Ly6C+ monocytes, Ly6G+ neutrophils, CD64+ macrophages, and NK1.1+ natural killer cells), and adaptive immune cells (C) (CD19+ B cells, CD4+ T cells, and CD8+ T cells). $P$-values represent the Mann-Whitney non-parametric $t$-test. ** = $P < 0.01$; ns = non-significant.

centrifuged at 107,240 × $g$ for 18 h or 144,302 × $g$ for 5 h. Virions were collected and dialyzed against dialysis buffer (150 mM NaCl, 15 mM MgCl, and 10 mM Tris-HCl [pH 7.4]). Viral titers were determined by plaque assay using L929 fibroblasts. To confirm the viral gene segments, viral dsRNA was purified from virions and separated by SDS-PAGE. Gene segments were visualized by ethidium bromide staining (9).

## Mouse experiments

C57BL/6 mice were used. Neonatal mice (4 to 5 days old) were mock infected orally with PBS for the control groups or infected orally with $10^4$, $10^5$, $10^6$, and $10^7$ PFU of reovirus T3D. Infected mice were monitored for up to 35 days and sacrificed if moribund. Hearts were resected and analyzed.

## Histology and immunohistochemistry

Neonatal mice (4 to 5 days old) were mock -infected orally with PBS or infected orally with $10^4$, $10^5$, $10^6$, or $10^7$ PFU of T3D. At 4 weeks post-infection, mice were euthanized, and the hearts were resected. Resected hearts were fixed in 10% neutral buffered formalin overnight. The fixed resected hearts were submitted to the JHU Experimental Pathology Core Laboratory for paraffin embedding and sectioning (4 µm thickness). Cardiac sections were stained with Masson's trichrome and examined blindly on a Leica optical microscope with the assistance of a cardiac pathologist. Bright field images were acquired using a bright field scanner using a 40× magnification and uploaded to Proscia (Philadelphia, PA, USA). Digital images were further enlarged to optimize visualization, and error bars were added using Proscia software.

## Echocardiogram

Neonatal mice (4 to 5 days old) were mock infected orally with PBS or infected orally with $10^4$, $10^5$, $10^6$, or $10^7$ PFU of T3D. M-mode echocardiography was performed on conscious mice between 28 and 35 days post-infection using a VisualSonics Vevo 2100 Imaging System. M-mode parasternal short images were obtained. Heart rate, systolic and diastolic diameters, systolic and diastolic LV volume, SV, EF, fractional shortening,

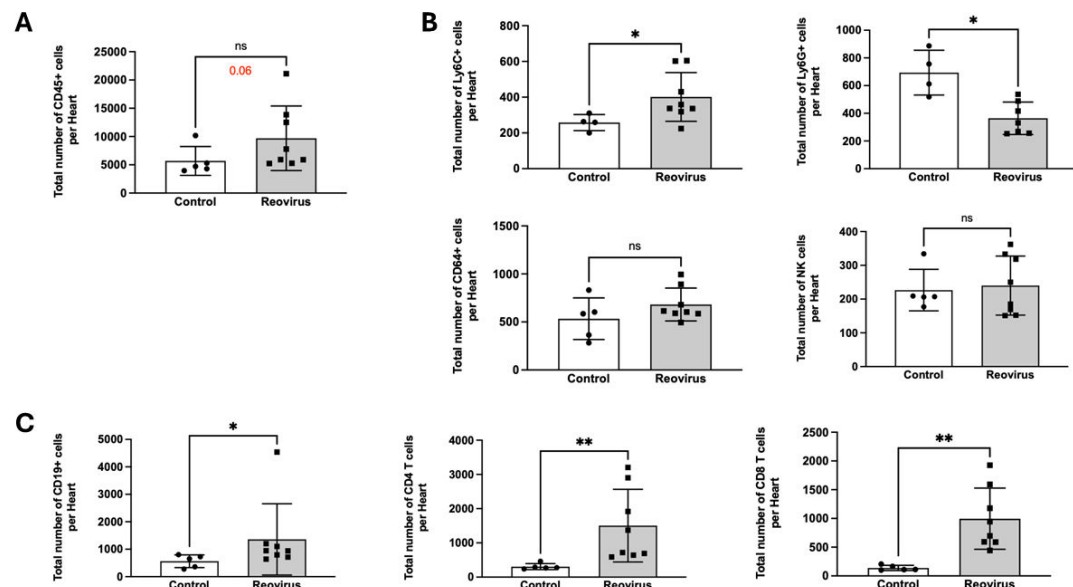

**FIG 4** T3D-induced myocardial immune infiltrate persists at 4 weeks post-infection. Neonatal C57BL/6 mice were mock-infected (Control; *N* = 5) or infected orally with 10$^6$ PFU of T3D (*N* = 8). At 26 days post-infection, the total number of cells per heart was quantified via flow cytometry focusing on the following cell types: (A) CD45+ cells, (B) innate immune cells (Ly6C+ monocytes, Ly6G+ neutrophils, CD64+ macrophages, and NK1.1+ natural killer cells), and (C) adaptive immune cells (CD19+ B cells, CD4+ T cells, and CD8+ T cells) were quantified by flow cytometry. *P*-values represent the Mann-Whitney non-parametric *t*-test.* = *P* < 0.05; ** = *P* < 0.01; ns = non-significant).

cardiac output, LV mass, systolic and diastolic LV anterior wall, and systolic and diastolic LV posterior wall were measured using Vevo Lab Software v5.8.2.

## Flow cytometry

Neonatal mice (4 to 5 days old) were mock infected orally with PBS or infected orally with T3D. At indicated times (8 or 26 days post-infection), mice were euthanized and hearts were perfused with 3 mL of Hanks' balanced salt solution (HBSS) and resected. Hearts from individual mice were digested with a buffer containing 50 U/mL DNase I (Sigma), collagenase II (Sigma), and 20 U/mL hyaluronidase (Sigma). Cells were washed with 7 mL of HBSS, and red blood cells were lysed with 2.5 mL ACK lysing buffer (Thermo Fisher). To generate a single-cell suspension, digested hearts were passed through a 40 µm filter (Fisher Scientific). Heart cells were transferred into a FACS tube (Life Sciences) for staining. Cells were washed in FACS buffer (1× PBS, 10 mL heat-inactivated FBS, and 2.0 mL EDTA 500 mM), and Fc receptors were blocked with anti-mouse CD16/CD32 (Biolegend). Cells were surface stained using Zombie Aqua-BV480 (Biolegend) as well as CD45-PerCP/Cy5.5 (Biolegend), CD8-BV785 (Biolegend), Ly6C-BV650 (Biolegend), CD19-APC (Biolegend), CD64-PE Cy7 (Biolegend), Ly6G-APC Cy7 (Biolegend), CD4-BUV737 (Biolegend), and NK-PE (Biolegend). Following surface staining, samples were fixed in 2% PFA (Thermo Scientific). Fluorescence minus one controls were used to set positive gates. Sample acquisition was performed on a BD FACSymphony flow cytometer (BD Biosciences) and analyzed using FlowJo software (BD). Each digested heart was analyzed in full, running the whole digested heart through the flow cytometer. The total number of events collected within each gate is reported. The number of cells per heart was approximately 50,000 CD45+/single alive cells/heart at day 8 post-infection and approximately 8,000 CD45+/single alive cells/heart at day 26 post-infection. Analysis was performed gating first on alive cells, then on single cells, and then on CD45+ cells. To identify myelomonocytes, we first gated CD11b+ cells. Within CD11b+ cells, we identified Ly6G+ cells. CD11b+ Ly6G− cells were further gated on a gate with Ly6C vs CD64 to identify Ly6C+ cells and CD64+ cells. CD45+ cells were gated against

CD19 to identify B cells. CD45+ CD11b− cells were plotted against CD4, CD8, NK 1.1 to identify T cells and NK cells. Representative gating strategy is reported in Fig. S2.

## Statistical analysis

Statistical analysis was performed using Prism software (GraphPad Software Inc.). Differences in total cell number were evaluated using the Mann-Whitney non-parametric $t$-test. Differences in echocardiography-based cardiac parameters and differences in the prevalence of histological lesions were determined using the one-way analysis of variance (ANOVA) followed by pair-wise comparisons. We used the false discovery rate method to adjust for multiple comparisons. $P$-values less than 0.05 were considered statistically significant.

## ACKNOWLEDGMENTS

This study was funded through NHLBI grants 5K08HLO145108 and 1R01HL160716 to L.A. and T32-HL007227 to M.E.F.A. and institutional funds from the Johns Hopkins Division of Cardiology awarded to L.A.

## AUTHOR AFFILIATIONS

[1]Division of Cardiology, Department of Medicine, Johns Hopkins University School of Medicine, Baltimore, Maryland, USA

[2]Department of Pathology, Johns Hopkins University School of Medicine, Baltimore, Maryland, USA

[3]Department of Microbiology and Immunology, University of Arkansas for Medical Sciences, Little Rock, Arkansas, USA

## AUTHOR ORCIDs

Kshipra S. Keole http://orcid.org/0009-0001-3867-9501
Karl W. Boehme http://orcid.org/0000-0003-2691-9763
Luigi Adamo http://orcid.org/0000-0003-2704-978X

## AUTHOR CONTRIBUTIONS

Maryam Ebadi Fard Azar, Formal analysis, Writing – original draft | Marcelle Dina Zita, Conceptualization, Formal analysis, Writing – original draft | Kshipra S. Keole, Writing – original draft | Sylvie Rousseau, Writing – original draft | Charles Cohen, Writing – original draft | Dao-Fu Dai, Writing – original draft | Karl W. Boehme, Conceptualization, Supervision, Writing – original draft, Writing – review and editing | Luigi Adamo, Conceptualization, Supervision, Writing – original draft, Writing – review and editing

## ETHICS APPROVAL

Animal husbandry and housing were performed following the guidelines of the Johns Hopkins University Animal Care and Use Committee at the Johns Hopkins School of Medicine.

## ADDITIONAL FILES

The following material is available online.

### Supplemental Material

**Fig. S1 (Spectrum02108-25-s0001.TIF).** Echocardiography analysis at 4 weeks post-infection.
**Fig. S2 (Spectrum02108-25-s0002.TIF).** Flow cytometry gating strategy.
**Supplemental material (Spectrum02108-25-s0003.docx).** Supplemental figure legends.

## Open Peer Review

**PEER REVIEW HISTORY (review-history.pdf).** An accounting of the reviewer comments and feedback.

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
