## [Reviewer comments · Microbiology Spectrum]

Microbiology Spectrum

Long-term effects of reovirus strain T3D on the myocardium

Maryam Ebadi Fard Azar, Marcelle Zita, Kshipra Keole, Sylvie Rousseau, Charles Cohen, Dao-Fu Dai, Karl Boehme, and Luigi Adamo

Corresponding Author(s): Luigi Adamo, Johns Hopkins University School of Medicine

Review Timeline:

Submission Date:	July 17, 2025
Editorial Decision:	August 6, 2025
Revision Received:	September 16, 2025
Editorial Decision:	November 3, 2025
Revision Received:	November 24, 2025
Accepted:	November 27, 2025

Editor: Gabriel Parra

Reviewer(s): The reviewers have opted to remain anonymous.

Transaction Report:

DOI: <https://doi.org/10.1128/spectrum.02108-25>

Re: Spectrum02108-25 (Long-term effects of reovirus strain T3D on the myocardium)

Dear Dr. Luigi Adamo:

Thank you for submitting your manuscript to Microbiology Spectrum. Your manuscript was reviewed by two subject matter experts. Both recognized the value of your study and recommended a few modifications and clarifications, as outlined below and attached to this email.

Revision Guidelines

Sincerely,
Gabriel Parra
Editor
Microbiology Spectrum

Reviewer #1 (Comments for the Author):

This manuscript by Azar et al. reports the development of chronic subclinical myocarditis in 26- to 35-day-old C57BL/6 mice infected as neonates (day 4) perorally with Type 3 Dearing reovirus. The authors find thinning of the LVAW day 26-35 in mice infected with 10^4 PFU orally, but not with higher doses. They found minimal evidence of changes in heart function but histological evidence of cardiac fibrosis with infiltration of adaptive immune cells. The authors suggest that ongoing chronic inflammation indicates the persistence of reovirus antigen. Overall, this is a well-written paper. The experiments and results are clear. The findings are of modest interest, but the potential safety concerns to humans will be of interest to those using

reoviruses as oncolytic agents. I have a few minor corrections and concerns:

1) The authors suggest that chronic inflammation in the hearts of the infected mice is a consequence of reovirus antigen persistence (lines 129-130). However, no evidence is presented to justify this speculation. It would also seem possible that the immune infiltration is a consequence of a response to self-antigens, as has been reported for coxsackievirus. Perhaps a sentence to indicate that the underlying reason for the persistent immune response is not known. Reovirus infection of the biliary tract has been associated with autoimmune disease; therefore, it is possible that the same phenomenon occurs in the heart.

2) Line 116-117. The sentence "Adaptive immune cells, B cells (CD19+), CD4 T cells, and CD8 T cells (Fig. 3C)." does not have a verb.

3) Line 215 'xx magnification': the magnification should be stated.

Reviewer #2 (Comments for the Author):

PLease see attached review document.

Title: Long-term effects of reovirus strain T3D on the myocardium

Authors: Maryam Ebadi Fard Azar, Marcelle Dina Zita, Sylvie Rousseau, Charles Cohen, Kshipra SKeole, Dao-Fu Dai, Karl W. Boehme and Luigi Adamo

This manuscript describes studies to explore the effects of reovirus T3D oral infection on the myocardium of neonatal mice, using immunohistochemistry, flow cytometry and EEG studies. The effects of reovirus on the myocardia have been described using T1L and reassorted viruses combining T1L and T3D, which result in more severe myocarditis than is observed in T1L infection. This work adds to the body of published work by exploring the potential of T3D reovirus in the myocardium. This virus model is more typically used for encephalitis models as it is known to transmit to the CNS via neuronal transmission, therefore studying the myocardium in neonates in this model appears novel.

General Comments:

In all presentation of the data, it is unclear how many animals are used in these studies. Group sizes are not provided in the Methods section, Results section or the Figure legends. The author should provide the group sizes for all studies to ensure sufficient numbers are provided for the statistics given in the manuscript. These data are largely not statistically significant, though there are trends towards an increase in lesions and inflammation in the CNS. It is unclear whether a larger study group would improve the significance of these studies as the number of animals is not provided and the number of trials is not described.

Methods:

More details regarding the ECG method in this study could be provided. How were the animals prepared, the length of recording time, the instrumentation and conditions used for the tests. These conditions should be consistent between reads and between groups to reduce variability of the readings – as such they should be provided.

Flow cytometry. FMO controls are described in this section. Appropriate compensation controls are not described in this section. The number of events collected and the number of cells isolated from the neonatal hearts of mice are not provided, though total number must be known to calculate total number in data provided.

Figures:

Figure 1 and supplementary Figure 1: number of mice per group undergoing recordings is not presented. What the dots in the charts represent is not described either. Is each dot a mouse or an individual recording? How many trials were performed? How many recordings per trial?

Figure #2:

No uninfected control images are provided for comparison to the infected control slices. Notations should be added to the imaging providing: 1. Scale bar, 2. Demarcation of the lesions described for reader, 3. The location of the lesions in the heart (ventricular or atrial walls) should be described. Images of lower magnification showing the density of lesions would also be beneficial. Also, what the dots in the quantitative chart (A) are indicating is not described. Is each dot a mouse, or a lesion?

The methods described H&E staining that was performed on these slices. Unfortunately, these data were not presented here. Such H&E staining would support the finding of infiltrating cells found by flow cytometry and indicate the location of the infiltrating cells relative to the lesions. As such H&E staining would be highly supportive and should be included here. These data may also provide a link between inflammation and lesion formation.

Figure legend indicates "XX" magnification. This should be updated with appropriate magnification, but scale bars on the imaging is still absolutely required. How the images were captured and digitally prepared should be described in the methods.

Figure 3 and 4:

These figures quantify the flow cytometry from both early and late time points following infection with T3D. It is not surprising that primarily innate cells are observed at d8 dpi with a larger acquired component found at 26 dpi. It is not made clear how many cells these data represent from each host. A presentation of the gating strategy for these experiments and representative flow plots for these quantitative data are not provided. The flow cytometry data and percentage of each cell type should be provided at least as supplementary data to allow full interpretation of the presented numbers and to support the conclusions made.

In addition to these modifications in the data, this reviewer would also suggest providing data about the presence of absence of reovirus in or near the heart. Evidence of successful inoculation, including neuronal infection following oral administration in a dose-dependent manner would support the model described. Further, one could ask whether the cardiac nerves or the thoracic ganglia are a potential reservoir for infection. Immunohistochemistry or in situ hybridization for reovirus in the myocardium and the nervous system would be beneficial. Alternatively, nested quantitative PCR for viral RNA could indicate low level infection of the myocardium.

We appreciate the reviewers' comments and have revised our manuscript accordingly. We think that the revisions we have made based on their feedback significantly enhance the quality of our paper.

In response to the specific comments of reviewer #1:

1. The authors suggest that chronic inflammation in the hearts of the infected mice is a consequence of reovirus antigen persistence (lines 129-130). However, no evidence is presented to justify this speculation. It would also seem possible that the immune infiltration is a consequence of a response to self-antigens, as has been reported for coxsackievirus. Perhaps a sentence to indicate that the underlying reason for the persistent immune response is not known. Reovirus infection of the biliary tract has been associated with autoimmune disease; therefore, it is possible that the same phenomenon occurs in the heart.

We thank the reviewer for the suggestion. We have modified the text on lines 131-136 of the revised manuscript to account for the reviewer's comment.

2. Line 116-117. The sentence "Adaptive immune cells, B cells (CD19+), CD4 T cells, and CD8 T cells (Fig. 3C)." does not have a verb.

We edited the sentence on lines 115-117 of the revised manuscript.

3. Line 215 'xx magnification': the magnification should be stated.

We apologize for this typo. Detailed information about image acquisition has been added at line 166-169.

In response to the specific comments of reviewer #2:

1. In all presentation of the data, it is unclear how many animals are used in these studies. Group sizes are not provided in the Methods section, Results section or the Figure legends. The author should provide the group sizes for all studies to ensure sufficient numbers are provided for the statistics given in the manuscript. These data are largely not statistically significant, though there are trends towards an increase in lesions and inflammation in the CNS. It is unclear whether a larger study group would improve the significance of these studies as the number of animals is not provided and the number of trials is not described.

The number of mice used for each experiment has been added to the figure legends.

2. More details regarding the ECG method in this study could be provide. How were the animals prepared, the length of recording time, the instrumentation and conditions used for the tests. These conditions should be consistent between reads and between groups to reduce variability of the readings –as such they should be provided.

We have provided additional details for the echocardiogram on lines 169-171 of the revised manuscript.

3. Flow cytometry. FMO controls are described in this section. Appropriate compensation controls are not described in this section. The number of events collected and the number of cells isolated from the neonatal hearts of mice are not provided, though total number must be known to calculate total number in data provided.

We have provided additional details for the flow cytometry experiments on lines 193-201 of the revised manuscript.

4. Figure 1 and supplementary Figure 1: number of mice per group undergoing recordings is not presented. What the dots in the charts represent is not described either. Is each dot a mouse or an individual recording? How many trials were performed? How many recordings per trial?

The number of mice used for each experiment has been added to the figure legends. We have provided additional details for the echocardiographic measurements (recordings) on lines 172-173, 218-219 and 232-233 of the revised manuscript.

5. Figure #2: No uninfected control images are provided for comparison to the infected control slices. Notations should be added to the imaging providing: 1. Scale bar, 2. Demarcation of the lesions described for reader, 3. The location of the lesions in the heart (ventricular or atrial walls) should be described. Images of lower magnification showing the density of lesions would also be beneficial. Also, what the dots in the quantitative chart (A) are indicating is not described. Is each dot a mouse, or a lesion?

Detailed information about image acquisition has been added at line 166-169. . We have edited the figure legend (Lines 232-233) to clarify that the dots in the plots represent the number of lesions per mouse and that a midventricular section was analyzed for each mouse.

6. The methods described H&E staining that was performed on these slices. Unfortunately, these data were not presented here. Such H&E staining would support the finding of infiltrating cells found by flow cytometry and indicate the location of the infiltrating cells relative to the lesions. As such H&E staining would be highly supportive and should be included here. These data may also provide a link between inflammation and lesion formation.

We apologize for this typo. H&E staining was not performed on all sections because Trichrome staining provided both staining of nuclei/cytoplasm obtained with H&E staining and staining of fibrous tissue. The reference to H&E has been removed from the manuscript.

7. Figure legend indicates “XX” magnification. This should be updated with appropriate magnification, but scale bars on the imaging is still absolutely required. How the images were captured and digitally prepared should be described in the methods.

Detailed information about image acquisition has been added at line 166-169. Scale bars are now more prominent and are referenced in the figure legend.

8. Figure 3 and 4: These figures quantify the flow cytometry from both early and late time points following infection with T3D. It is not surprising that primarily innate cells are observed at d8 dpi with a larger acquired component found at 26 dpi. It is not made clear how many cells these data represent from each host. A presentation of the gating strategy for these experiments and representative flow plots for these quantitative data are not provided. The flow cytometry data and percentage of each cell type should be provided at least as supplementary data to allow full interpretation of the presented numbers and to support the conclusions made.

The gating strategy is now described in detail within the methods section, lines 191-198. In this text, we clarify that we acquired all events via flow cytometry in each digested heart. We have also edited the pertinent figure legend to explain that the plots report the number of events per heart. A new supplementary figure (supplementary figure 2) reporting the gating strategy has also been added.

9. In addition to these modifications in the data, this reviewer would also suggest providing data about the presence or absence of reovirus in or near the heart. Evidence of successful inoculation, including neuronal infection following oral administration in a dose-dependent manner would support the model described. Further, one could ask whether the cardiac nerves or the thoracic ganglia are a potential reservoir for infection. Immunohistochemistry or in situ hybridization for reovirus in the myocardium and the nervous system would be beneficial. Alternatively, nested quantitative PCR for viral RNA could indicate low level infection of the myocardium.

We appreciated the reviewer’s suggestion. While examining reovirus presence in cardiac neural structures would provide important mechanistic insights about how the virus disseminates to the heart, these experiments are beyond the scope of the current project, as it focuses on the immune response to reovirus in the heart.

Re: Spectrum02108-25R1 (Long-term effects of reovirus strain T3D on the myocardium)

Dear Dr. Luigi Adamo:

Thank you for submitting your manuscript to Microbiology Spectrum. Reviewer number 2 is still requesting some minor modifications. Please address those comments and return the manuscript within 60 days; if you cannot complete the modification within this time period, please contact me. If you do not wish to modify the manuscript and prefer to submit it to another journal, notify me immediately so that the manuscript may be formally withdrawn from consideration by Spectrum.

Revision Guidelines

Sincerely,
Gabriel Parra
Editor
Microbiology Spectrum

Reviewer #2 Comments:

Most comments have been addressed with this revision. Please see additional minor comments below.

Figure 2: Additional Panel showing Mock-infected control tissue for reference needs to be added for comparison. A lower magnification (wider field) image of the region showing lesions in context would be beneficial. Using arrows, mark immune cells and/or increased fibrosis in tissues to help readers interpret images.

Supplementary Figure 2:

No legend provided. Average number of Live cells from hearts should be included (in legend or methods). Reported number(s) start at CD45+ which is less than 1 % of the total heart population collected by this figure. Was this sample taken at (representative to) the early (8d) or late time (d26) point? Please be consistent in the type of plot used in the figure - all dot plot

or all density plot.

Most comments have been addressed with this revision.

Please see minor comments below.

Minor comments:

Figure 2: Additional Panel showing Mock-infected control tissue for reference needs to be added for comparison. A lower magnification (wider field) image of the region showing lesions in context would be beneficial. Using arrows, mark immune cells and/or increased fibrosis in tissues to help readers interpret images.

Supplementary Figure 2:

No legend provided. Average number of Live cells from hearts should be included (in legend or methods). Reported number(s) start at CD45+ which is less than 1 % of the total heart population collected by this figure. Was this sample taken at (representative to) the early (8d) or late time (d26) point? Please be consistent in the type of plot used in the figure – all dot plot or all density plot.

Editorial comments:

1. Bright field should be “brightfield”.
2. Scale bars in Figure 2 would be clearer on the right side of the image, with no white background. With 40 x, a 50 um scale bar might be cleaner and more informative
3. Supplementary Figure 1 legend “ 10^7 (N=)”. Please add number of N.
4. Line 234: “mid papillary”, should be same as “mid-papillary” as in previous sentence.
5. No legend for supplementary figure 2 is provided.

We appreciate the reviewers' comments and have revised our manuscript accordingly. We think that the revisions we have made based on their feedback significantly enhance the quality of our paper.

In response to the specific comments of reviewer #2:

1. Figure 2: Additional Panel showing Mock-infected control tissue for reference needs to be added for comparison. A lower magnification (wider field) image of the region showing lesions in context would be beneficial. Using arrows, mark immune cells and/or increased fibrosis in tissues to help readers interpret images.

We thank the reviewer for the suggestion. We have added the image with the arrows depicting the fibrosis as well as a representative control image.

2. Supplementary Figure 2:

No legend provided. Average number of Live cells from hearts should be included (in legend or methods). Reported number(s) start at CD45+ which is less than 1 % of the total heart population collected by this figure. Was this sample taken at (representative to) the early (8d) or late time (d26) point? Please be consistent in the type of plot used in the figure - all dot plot or all density plot.

In the revised manuscript, we have added a legend for Supplementary Figure 2. In addition we have specified the average number of live, single, CD45+ cells per heart within the methods section. We have specified that the representative plots reported in Supplementary Figure 2 correspond to a late time point.

The plots in this supplementary figure are depicted differently because each panel represents a sequential gating step, not the same population. Each gate refines the dataset, moving from all cardiac cells to specific immune subsets (e.g., CD45+ leukocytes, CD11b+ myeloid cells, CD4+/CD8+ T cells, etc.). Thus, the cell distribution naturally changes at each step, and identical plots would misrepresent the gating hierarchy and underlying biology.

Re: Spectrum02108-25R2 (Long-term effects of reovirus strain T3D on the myocardium)

Dear Dr. Luigi Adamo:

Thank you for submitting your revised manuscript. The manuscript has been now accepted, and I am forwarding it to the ASM production staff for publication. Your paper will first be checked to make sure all elements meet the technical requirements. ASM staff will contact you if anything needs to be revised before copyediting and production can begin. Otherwise, you will be notified when your proofs are ready to be viewed.

Sincerely,
Gabriel Parra
Editor
Microbiology Spectrum